# Remote Sensing Monitoring of Vegetation Reclamation in the Antaibao Open-Pit Mine

**Jiameng Hu [1], Baoying Ye [1,2], Zhongke Bai [1,2,3,*] and Yu Feng [1]**

1 School of Land Science and Technology, China University of Geosciences, Beijing 100083, China
2 Key Lab of Land Consolidation and Rehabilitation, The Ministry of Natural Resources, Beijing 100035, China
3 Technology Innovation Center of Ecological Restoration Engineering in Mining Area, The Ministry of Natural Resources, Beijing 100083, China
* Correspondence: baizk@cugb.edu.cn

**Abstract:** After the regreening of the open-pit mine dump, vegetation usually needs to be managed and protected manually for several years before it reaches stability. Due to the spontaneous combustion of coal gangue, surface collapse, and other reasons, secondary damage may occur at any time. Regreening monitoring plays a vital role in the restoration and reconstruction of the mining ecosystem and can provide support for the timely replenishment of seedlings in the damaged area. In this study, remote sensing images were collected from 1986 to 2020 to obtain the NDVI distribution of dumps in the Antaibao open-pit coal mine. In order to obtain the overall growth law of regreening vegetation over time, the study adopted the unary regression analysis method and tested the correlation between NDVI and time by the Pearson correlation coefficient. However, through the Sen+Mann–Kendall trend analysis, it was found that there were differences in the trends of NDVI within the same dump. Next, by means of the Mann–Kendall mutation test and interactive interpretation, information, such as stable nodes of different regreening vegetation and vegetation growth patterns in degraded areas, were obtained. Through the above methods, the following conclusions were drawn: (1) The earlier the dumps were regreened, the more the areas were covered by significantly improved vegetation. In this study: 97.31% (the proportion of significantly improved vegetation in the south dump) >95.58% (the proportion in the west dump) >86.56% (the proportion in the inner dump) >79.89% (the proportion in the west expansion dump). (2) Different vegetation types have different time nodes for reaching stability. It takes about three years for wood, shrub, and a mix of grass, shrub, and wood to reach stability, but only one year for grass. (3) The destruction in mining areas is expansive and repeatable. Monitoring the growth patterns of regreening vegetation is conducive to understanding the reclamation effect, and provides a scientific basis for land reclamation planning and land management policies in the mining area. At the same time, the trend analysis method in this study can quickly extract problem areas after dump regreening and is applicable in most dumps.

**Keywords:** regreening monitoring; NDVI; unary regression analysis; Sen+Mann–Kendall trend analysis; Mann–Kendall mutation test; interactive interpretation

## 1. Introduction

According to the *Coal Industry Development Annual Report* in 2021 issued by the China National Coal Association, China's raw coal output reached 41.3 billion tons in 2021, an increase of 5.7% over the previous year. Moreover, coal will remain an important energy supply in the future [1]. However, while coal brings huge social resources, it also causes serious eco-environmental issues [2–4]. In particular, in the long-term coal mining process, the rock formation and soil above the coal seam are continuously peeled off, and the exfoliated materials accumulate to form massive dumps. According to statistics, the area generated by compaction is about 1.5–2.5 times the area damaged by excavation [5]. Large-scale land occupation destroys the original soil structure and requires soil reconstruction.

Soil reconstruction is not only a prerequisite for vegetation reclamation but also a key link in ecosystem restoration [6]. Relevant studies have pointed out that there is an interaction between vegetation and soil. On the one hand, regreening vegetation can reduce soil loss and improve soil functions, such as water holding capacity and infiltration capacity. On the other hand, these soil functions can enhance ecosystem services [7,8]. In the process of ecological restoration in mining areas, it is usually necessary to monitor the conditions of vegetation, soil, and other elements simultaneously to reflect the restoration effect.

Remote sensing technology has the advantages of providing large-scale, multi-temporal, and multi-angle feature information [9], and can provide technical support for comprehensive and rapid monitoring of the mining environment. With the progress of mining, the environment in the mining area will undergo complex changes. Due to the vulnerability of such environments, it is imperative to monitor the regreening trend during and after the restoration process. In recent years, remote sensing technology has been used more widely in the monitoring and evaluation of the mining ecosystem [10,11]. Some scholars monitored ecological changes in mining areas through remote sensing data, including surface collapse, climate difference, and topographic changes caused by mining [12]. Moreover, some surface information is obtained through the index method, such as normalized difference built-up index (NDBI), bare soil index (BSI), normalized burn ratio (NBR), etc. [13]. In addition, some serious pollution problems caused by mining activities are also monitored and quantified by means of remote sensing. The monitoring of soil pollution includes monitoring of soil heavy metal content, organic matter content, pH, and other indicators [14–16]. The pollution of water and air is mainly inverted by the combination of band information to quickly extract the pollution changes [17]. Moreover, some scholars used the index method and biomass method to extract the vegetation information and quantify the vegetation coverage, chlorophyll content, nitrogen content, etc., to characterize the changes in the ecological environment [18,19].

Judging from the existing research results, the application of remote sensing technology in mining areas is mostly concentrated in land cover change monitoring and environmental monitoring. However, long-term follow-up monitoring for reclamation vegetation is deficient. After the regreening project of the open-pit mine dump, vegetation may degrade again at any time due to the unsuitable climate, spontaneous combustion of coal gangue, or other reasons. Therefore, it is necessary to monitor the vegetation growth status during recovery for evaluating the reclamation effect [20]. In order to reveal the process characteristics of vegetation restoration in the mining area, this study took the Antaibao open-pit coal mine as an example, used the Landsat and HJ series remote sensing images from 1986 to 2020, and applied the unary regression analysis method to summarize the overall trend of regreen vegetation. Then the spatial heterogeneity was analyzed by the Sen+Mann–Kendall trend analysis.

The main objectives are as follows: (1) Based on the unary regression analysis, understanding the growth trends of different regreening vegetation overall, and comparing the differences in the restoration process will help to select suitable regreening vegetation species according to the reclamation plan. (2) By extracting areas whose trend is significantly worse than the surrounding area, the "problem areas" can be accurately extracted. Moreover, the problems that may occur in the restoration process can be found. By analyzing the change law of regreening vegetation, we can clearly know the reclamation effect of the Antaibao open-pit coal mine and will provide a theoretical basis for improving ecological monitoring and evaluation of land management in the mining area.

With the enhancement of the greenhouse effect, $CO_2$ emission reduction has become key research content for relevant scholars. Due to the strong interference of human activities in mining areas, the entire process from coal mining to post-mining reclamation is closely related to the carbon cycle. The reclaimed vegetation, as the main surface carbon sink element in the mining area, plays an important role in the carbon cycle process after mining restoration. This study is of great significance for understanding the carbon cycle process in the mining area by monitoring the long-term change trend of the regreening vegetation and tracking the vegetation restoration after the mining area reclamation. Furthermore, the results will provide a scientific basis for the formulation of land reclamation planning and policies.

## 2. Materials and Methods

### 2.1. Study Area

Antaibao open-pit coal mine is located in Shanxi Province, China, 112°10′–112°30′E and 39°23′–39°37′N, which belongs to the ecologically fragile area in the eastern of the Loess Plateau [21]. The climate here is typical temperate arid to a semi-arid continental monsoon climate. Moreover, the seasons are distinct, and are characterized by less rain and snow in spring, concentrated rainfall in summer, less rain in autumn, and more wind and less snow in winter. The average annual temperature ranges from 5.4 °C to 13.8 °C, and the total annual precipitation averages 426.7 mm, of which, 75–90% occurs in the rainy season [22].

Since the start of mining in 1985, the Antaibao open-pit mine has adopted integrated engineering technology and reclamation while mining [23]. According to the engineering sequence of "stripping-mining-reclamation", the circulation mode of "Excavation-Transportation-Dumping-Reshaping-Reclamation" has formed [24]. The production scale of the Antaibao open-pit coal mine is relatively large, and the mining time is relatively long. With the continuous development of mining, various land use types, such as mining pits, stripping areas, industrial sites, and dumps are formed in turn. According to Figure 1, the Antaibao coal mine, together with the adjacent Nansigou and Anjialing mines, formed the most modernized Pingshuo coal base in China, covering an area of about 380 km². To date, the four dumps in the Antaibao mining area have been fully regreened. Firstly, the south dump began to be reclaimed around 1987, and the regreening vegetation cover type is mainly wood, including black locust (*Robinia pseudoacacia* Linn.), sea buckthorn (*Hippophae rhamnoides* Linn.), caragana (*Caragana korshinskii* Kom), and poplar (*Populus* L.). However, due to the spontaneous combustion of coal gangue, the surface temperature rose, and vegetation in some areas was repeatedly destroyed, which severely slowed down the recovery of the ecosystem. Then the west dump was reclaimed in 1989, with the goal of increasing vegetation coverage density, and currently, it is mainly covered by sea buckthorn shrubs. Moreover, the reclamation of the west expansion dump started in 2008, and is currently mainly covered by grasses, including alfalfa (*Medicago sativa* L.) and erect milkvetch (*Astragalus adsurgens* pall.). Finally, the inner dump began to be reclaimed in 1998, and the reclamation work is expected to continue until 2059, and vegetation type is mainly combined grass, shrub, and wood [25,26], covered by sea buckthorn, narrow-leaved oleaster (*Elaeagnus angustifolia* Linn.), elm (*Ulmus pumila* L.), and caragana. The location of the study area and each dump site is shown in Figure 1.

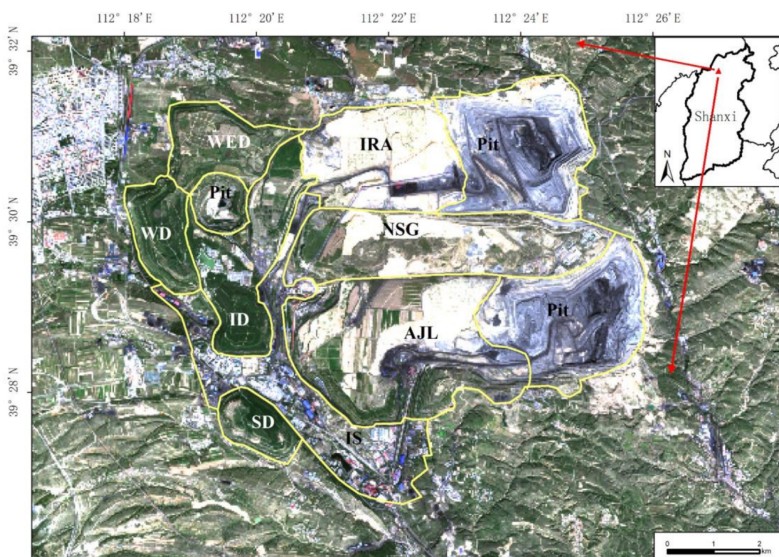

**Figure 1.** Location of the study area. SD, WD, WDE, and ID are, respectively, the abbreviations for the south dump, the west dump, the west expansion dump, and the inner dump of the Antaibao mine. IRA, NSG, AJL, and IS are abbreviations for the initial reclamation area of the inner dump, Nansigou, Anjialing mine, and industrial sites, respectively. The remote sensing image was from Sentinel-2 remote sensing data, 17 September 2020.

### 2.2. Data Collection

Landsat remote sensing images were all Landsat Collection 1 data, obtained from the National Aeronautics and Space Administration (NASA) and downloaded from the USGS website (http://earthexplorer.usgs.gov/, accessed on 14 July 2021). Moreover, the processing levels of the data mainly involve terrain precision correction (L1TP). However, due to missing data in parts, the data for 1995 and 1998 was supplemented with the Geometric Systematic Correction (L1GS) data. The Landsat satellites are launched for the purpose of detecting earth resources. From the launch of the first satellite in 1972 to the launch of the Landsat 9 satellite in September 2021, nine Landsat satellites have been launched. Therefore, Landsat satellites have significant time scale advantages and are widely used in disaster monitoring, resource census, crop assessment, and climate change research, etc. [27–30]. However, due to the fault of the Landsat 7 ETM+ airborne Scan Line Corrector (SLC), the images acquired after 2003 lost band information, which seriously affected the quality of the Landsat 7 ETM+ images. The Landsat series data used in this study included Landsat 4/5 TM images from 1986–2011 and Landsat 8 OLI images from 2013–2020. Landsat 4 and Landsat 5 were, respectively, launched in 1982 and 1984, and both carried TM sensors. The sensor TM includes information for 7 bands, with a spatial resolution of 30 m and 120 m, and a scanning period of 16 days. Moreover, the TM sensor is in good working condition and has acquired earth images for many years. Its band information is used to distinguish vegetation, identify rock minerals, and sense thermal radiation. Landsat 8 carries an operational land imager (OLI) and thermal infrared sensor (TIRS), covering a total of 11 wavebands from thermal infrared to visible light. Among them, OLI can passively sense the solar radiation reflected by the surface, covering a total of 9 bands from infrared to visible light, and is mainly used for coastal observation, soil and vegetation discrimination, etc. TIRS includes two separate thermal infrared wavebands, mainly used to induce thermal radiation.

In order to ensure the continuity of sequence data, the data in 2012 is supplemented by environmental satellite (referred to as HJ) data with the same spatial resolution [25]. The HJ satellite is a China earth observation satellite for environmental and disaster monitoring, consisting of two optical satellites and one radar satellite. It is mainly used for large-scale dynamic monitoring of the ecological environment and disasters, timely reflecting the occurrence of a disaster, predicting the development and change trends of the ecological

environment, and quickly assessing the disaster situation. The HJ image was downloaded from the China Resources Satellite Application Center website (http://www.cresda.com/CN/, accessed on 22 July 2021). In addition, the images used in this study were all selected from June to September, which can effectively reflect the growth state of vegetation. The data information is as follows (Table 1).

**Table 1.** Data list includes information on remote sensing images applied in this study.

| Sensor Type | Spatial Resolution (m) | Year |
|---|---|---|
| Landsat 4/5 TM | 30 × 30 | 1986–2011 |
| HJ1B CCD2 | 30 × 30 | 2012 |
| Landsat 8 OLI | 30 × 30 | 2013–2020 |

*2.3. Methods*

In this study, the long-term regreening monitoring is mainly based on NDVI, using Landsat and HJ remote sensing images as the basic data sources, and obtaining the NDVI distribution from 1986 to 2020 through band calculation. Then, with the initial regreening time as the starting point, time series analysis was carried out on the four dumps respectively, including the unary regression analysis overall and the Sen+Mann–Kendall trend analysis on the spatial scale, to obtain the overall trends and differences in the spatial distribution of different regreening vegetation. Next, based on the results of the Sen+Mann–Kendall trend analysis, we extracted the "problem area" (the area where vegetation growth is significantly worse than the surroundings) and the "typical area" (area located in the middle of dump and reclaimed for more than 8 years, extracted for mutation test), and verified by interactive interpretation. Finally, the Mann–Kendall mutation test was performed on the "typical area" to obtain the time nodes when different vegetation types reached stability. The technical workflow for this study is shown below in Figure 2:

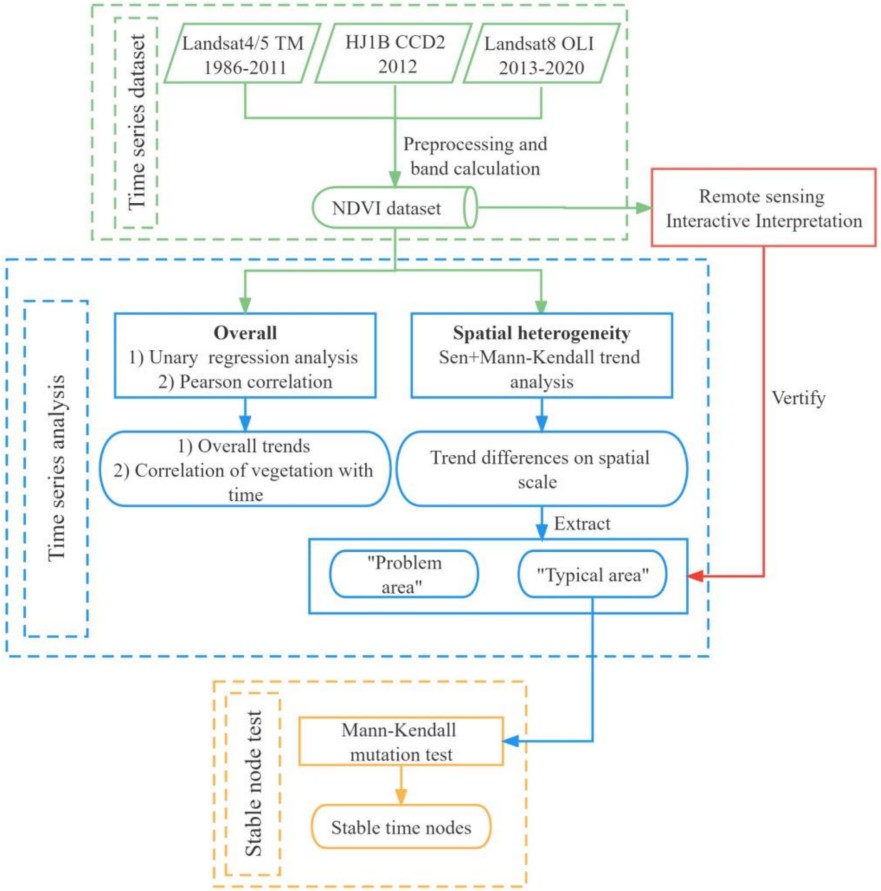

**Figure 2.** Overview of the technical workflow for this study.

### 2.3.1. Preprocessing and Calculation of NDVI

In the process of satellite sensor imaging, it will inevitably be affected by solar radiation, atmospheric scattering, etc., as well as by the structure and optical characteristics of the satellite sensor. In order to eliminate the internal error caused by sensor sensitivity and the external error caused by atmospheric scattering, the original image data needs to be preprocessed: radiometric calibration and atmospheric correction. In this study, the preprocessing of original image data is mainly completed by ENVI5.3 software. By radiometric calibration, the original DN value was converted into the radiance value of the pixel, which is convenient for further conversion in the later band calculation. Then, the atmospheric correction is realized through the FLAASH Atmospheric Correction function, to obtain the real surface reflectivity of the ground objects. The preprocessed images eliminated errors in the shooting process and would be used for the later vegetation characterization through the vegetation index method.

Among the existing research results, there are more than 100 kinds of vegetation indices [31–33], and the normalized difference vegetation index (NDVI) is the most widely used one, proposed by Rouse et al. in 1974 [34]. Compared with other vegetation indices, NDVI has the following advantages: (1) After the ratio processing of NDVI, the errors caused by the sun elevation angle, terrain slope, and satellite observation angle can be eliminated to a certain extent [35]; (2) it can eliminate the influence of earth atmosphere, such as the influence of ozone on the reflection of the red light band and the near-infrared band; (3) it can eliminate the influence of water and bare soil, and enhance the ability to characterize the growth state of vegetation. In conclusion, NDVI can reflect the growth state and distribution of vegetation macroscopically and is suitable for research on vegetation classification and monitoring, climate change monitoring, and ecological environment change monitoring [36–38].

### 2.3.2. Remote Sensing Interactive Interpretation

Interactive interpretation refers to the process in which the interpreter obtains information about the objects by directly observing images [39]. In this study, after selecting the "problem area", the visual interpretation method is used to verify the accuracy of the Sen+Mann–Kendall trend analysis results. Moreover, the methodology is based on the premise that mining and reclamation will cause abrupt changes to vegetation, which can be shown by vegetation indices. This study mainly used NDVI values as a datum reference to characterize the growth state of regreening vegetation. When the NDVI values were tracked, the spatial distribution differences in adjacent years were compared, and the areas with significant changes within two years were extracted. The main steps for extracting dynamic change information are as follows: First, arrange the NDVI sequence sets year by year, and compare the vegetation changes in adjacent years with the help of the mapping and display functions of ArcGIS 10.3. Then, based on the comparison results, the areas with abrupt changes were marked. This is because, under natural conditions, the vegetation growth will not change too drastically within two years. If the NDVI increases significantly in a short period of time, it means that human intervention has occurred during this period, thus confirming that there are land reclamation activities in this area. Finally, combined with the actual situation and the marking records, the reclamation area is determined for the plots where the vegetation trend has improved significantly, and the plots with significantly decreased NDVI are determined as the damaged area or the area where disturbance occurs.

### 2.3.3. Unary Regression Analysis

Unary regression analysis is a traditional statistical analysis method, widely used in data analysis and data prediction, including the reasonable interpretation of the prediction process and prediction results [40]. In order to obtain the change law of regreening vegetation over time, NDVI and regreening year were used as regression analysis variables to evaluate the linear relationship between the two. Moreover, in order to measure the

correlation between the regreening year and the vegetation index, the Pearson correlation coefficient is also used in Equation (4). In this study, it is assumed that the independent variable is $x$, which represents the order of the years, and the dependent variable is $y$, which represents the annual average of NDVI. Then the univariate linear regression model is shown as in Equation (1).

$$\hat{y} = \hat{a} + \hat{b}x \tag{1}$$

where the equation represents the fitted equation between the variables; $\hat{y}$ is the estimated value of $y$, also known as the regression value. According to the least square method, parameter $\hat{a}$ and $\hat{b}$ are shown in Equations (2) and (3).

$$\hat{a} = \overline{y} - \hat{b}\overline{x} \tag{2}$$

$$\hat{b} = \frac{n * \sum_{i=1}^{n} x_i y_i - \left(\sum_{i=1}^{n} x_i\right)\left(\sum_{i=1}^{n} y_i\right)}{n * \sum_{i=1}^{n} x_i^2 - \left(\sum_{i=1}^{n} x_i\right)^2} \tag{3}$$

$$r = \frac{\sum_{1}^{n} (x_i - \overline{x})(y_i - \overline{y})}{\sqrt{\sum_{1}^{n} (x_i - \overline{x})^2 \sum_{1}^{n} (y_i - \overline{y})^2}} \tag{4}$$

where $x_i$, $y_i$ are the actual values, $\overline{x}$, $\overline{y}$ are separately mean values of $x_i$ and $y_i$ ($i$ = 1, 2, 3, ... $n$), and $r$ is between $-1$ and 1.

The Pearson correlation coefficient ($r$) is close to 1, the positive correlation between the two, the closer to $-1$, the negative correlation between the two, and the closer to 0, the less correlation between the two. The overall trends of NDVI changing with time after regreening were obtained through the unary regression analysis.

### 2.3.4. Sen+Mann–Kendall Trend Analysis

The Sen+Mann–Kendall (Sen+MK) trend analysis method combines the Theil–Sen estimator and the Mann–Kendall test. Theil–Sen estimator is often used to calculate trend values, and can reduce the interference of outliers by calculating the median value of the sequence dataset. However, it cannot realize the judgment of the significance of the trend itself [41,42]. Therefore, it is necessary to combine the MK test, which can complete the test of the significance of the trend. The Mann–Kendall trend analysis is a nonparametric statistical method [43], also used for trend detection of precipitation and drought frequency under the influence of climate change, which can be used to classify data trends into insignificant and significant trends [44–46]. In this study, this method was used to test the significance of change trends for the regreening vegetation. First, the Sen trend value is calculated as in Equation (5).

The Sen trend value estimation:

$$beta = median\left(\frac{x_j - x_i}{j - i}\right), \; j > i \tag{5}$$

where the Sen trend value ($beta$) represents the changing trend of NDVI during the monitoring time, $i$ and $j$ represent the order of the years, $x_i$ and $x_j$ are the NDVI values of corresponding years. It should be noted that the calculation process in this study is carried out in units of pixels, and the results obtained are presented in the form of grids.

Then the MK test is used to judge the significance of the trend. The calculation process is divided into two parts: MK parameter calculation and bilateral test [47,48]. The calculations of related statistical values are shown in Equations (6)–(8).

Statistics $S$:

$$S = \sum_{i=1}^{n-1} \sum_{j=i+1}^{n} sgn(x_j - x_i), \text{ and } sgn(x_j - x_i) = \begin{cases} +1, & x_j > x_i \\ 0, & x_j = x_i \\ -1, & x_j < x_i \end{cases} \tag{6}$$

Variance $VAR(S)$:

$$VAR(S) = \frac{1}{18} \times n(n-1)(2n+5) \tag{7}$$

Statistics $Z$:

$$Z = \begin{cases} \frac{S-1}{\sqrt{VAR(S)}}, & S > 0 \\ 0 &, \quad S = 0 \\ \frac{S+1}{\sqrt{VAR(S)}}, & S < 0 \end{cases} \tag{8}$$

where $n$ is the number of time series, $S$ represents the change trend statistic. Moreover, $VAR(S)$ is the variance of the statistic $S$, and $Z$ represents significance.

The bilateral test of time series data requires that the time series meet certain conditions, the number of samples $n \geq 8$, to ensure that the statistic $S$ obeys the standard normal distribution. In order to ensure the validity of the time series analysis results, the sample size for time series analysis in this study is greater than 8. At a given significance level, the critical value found in the normal distribution table is $Z_{1-\alpha/2}$, and $1 - \alpha/2$ is the confidence interval. When $|Z| \leq Z_{1-\alpha/2}$, the null hypothesis is accepted, i.e., the trend is not significant; if $|Z| > Z_{1-\alpha/2}$, the null hypothesis is rejected, i.e., the trend is considered significant. In this study, the confidence level $\alpha = 0.05$, and $Z_{1-\alpha/2} = 1.96$. Based on this, the *beta* and $Z$ are reclassified as Table 2.

**Table 2.** Sen+MK trend analysis categories.

| beta | Z | Trend Features |
|---|---|---|
| beta > 0 | $\|Z\| > 1.96$ | Significant Increase |
|  | $\|Z\| < 1.96$ | Not Significant Increase |
| beta = 0 | Any value | No Change |
| beta < 0 | $\|Z\| < 1.96$ | Not Significant Decrease |
|  | $\|Z\| > 1.96$ | Significant Decrease |

2.3.5. Mann–Kendall Mutation Test

The Mann–Kendall mutation test is one of the most effective methods for testing time series mutations, which can identify the moment when the mutation begins and indicate the time period of the mutation [49,50]. In Equation (9), for a time series with $n$-year samples, the order sequence is first constructed $s_k$.

$$s_k = \sum_{i=1}^{k} r_i (k = 1, 2, \ldots, n), \text{and } r_i = \begin{cases} +1, & if\ x_i > x_j \\ 0, & else \end{cases} (j = 1, 2, \ldots, i) \tag{9}$$

where $r_i$ represents the annual average of NDVI, and $i, j$ represent the order of the years. Next, we calculate the statistics $E(s_k)$, $Var(s_k)$ as in Equations (10) and (11).

$$E(s_k) = \frac{n(n-1)}{4} \tag{10}$$

$$Var(s_k) = \frac{n(n-1)(2n+5)}{72} \tag{11}$$

where $E(s_k)$, $Var(s_k)$ are the mean and variance of $s_k$, respectively. Next, under the assumption that the time series is random and independent, define the statistics $UF_k$:

$$UF_k = \frac{s_k - E(s_k)}{\sqrt{Var(s_k)}} \tag{12}$$

After calculating $UF_k$ according to the above Equation (12), we repeat the above calculation process according to the reverse time sequence, and then take the inverse of the calculated value to obtain $UB_k$. We plot the curves of $UF_k$ and $UB_k$. If the intersection

of the two lines is within the confidence interval (−1.96, 1.96), the intersection is the mutation point.

## 3. Results

### 3.1. Monitoring Results in the South Dump

According to the images and historical records, it can be determined that the south dump began to be regreened in 1987, and it is the first dump in the Antaibao open-pit mine to be regreened. Taking the first regreening time as starting time, unary regression analysis overall and Sen+MK trend analysis on the spatial scale were carried out on the NDVI dataset. From the unary regression analysis results (Figure 3a), NDVI dataset has a positive correlation with time, i.e., with the increase of time, NDVI shows an increasing trend after regreening. Moreover, the Pearson correlation coefficient between NDVI and time reaches about 0.88, indicating that NDVI has a strong correlation with time. In addition, the annual average of NDVI showed a trend of rising and remained above 0.5 after 2002. From the perspective of trend analysis (Figure 3b), the NDVI sequence dataset shows an increasing trend overall, indicating that the improvement of vegetation is very significant after regreening. According to statistics, the area covered by significantly improved vegetation is 178.81 ha, accounting for about 97.31% of the south dump. In addition, there are two areas in the northwest and northeast where NDVI does not increase significantly, totally accounting for 1.86%. Moreover, at the southern edge, the local area shows a trend of a significant reduction in NDVI, which is confirmed to be caused by artificial illegal mining.

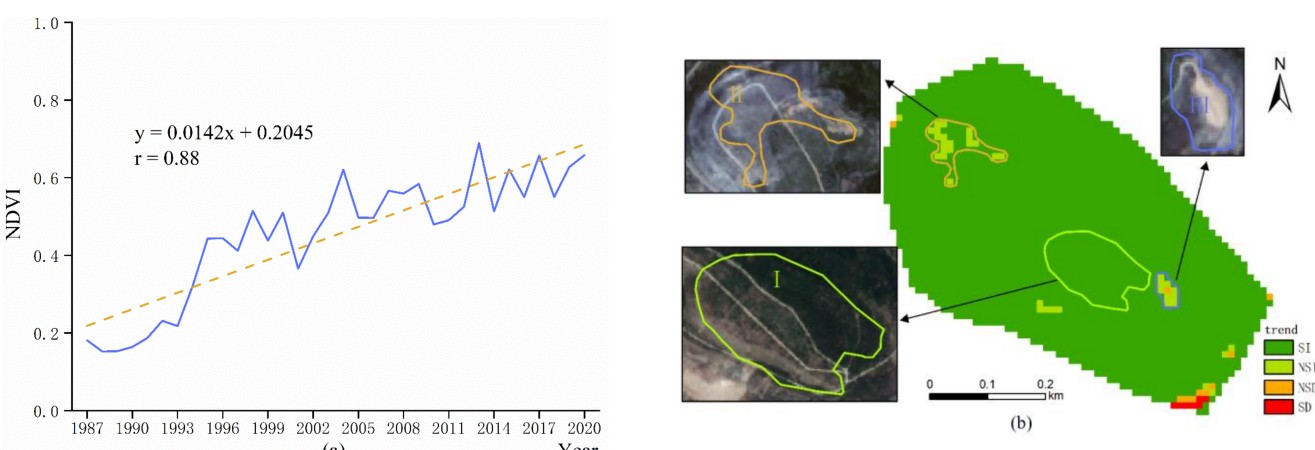

**Figure 3.** The time series analysis for NDVI on time and spatial scales in the south dump. (**a**) shows the results of the unary regression analysis. And (**b**) shows the results of the Sen+MK trend analysis. SI, NSI, NSD, and SD are, respectively, the abbreviations for significantly increase, not significantly increase, not significantly decrease, and significantly decrease.

The area with a notably different trend from the surroundings is defined as the "problem area". In addition, in order to obtain the time node for regreening vegetation reaching stability, the area with a significant increase in NDVI is defined as the "typical area". According to the Sen+MK trend analysis results, the "typical area" (area I) and the "problem areas" (areas II and III) were extracted, as shown in Figure 3b. Combined with interactive interpretation, it was found that all three areas began to regreen in 1994. In area I, NDVI showed a significant increasing trend, indicating that the regreening vegetation recovered steadily after regreening, and there was no excessive disturbance for area I. The regression analysis for area I can help to obtain the time node for regreening vegetation reaching stability under normal conditions. Area II and area III are located inside the dump, but the trends in these two areas are not obvious, indicating that the restoration of regreening vegetation is not significant. Moreover, in area III, the surroundings of

degraded vegetation showed a trend of insignificant increase, indicating that degradation is expansive.

After extracting the area range based on the Sen+MK trend analysis results, the year-by-year NDVI of each area was obtained by statistical methods and drawn into a line graph. According to Figure 4, NDVI in area I changed relatively large within 3 years after regreening, showing a trend of increasing firstly and then decreasing. Moreover, based on the Mann–Kendall mutation test method, the stable node of area I is three years. After more than 3 years, NDVI in area I still showed a continuous fluctuating trend, but the band amplitude was always relatively small. Moreover, after 2003, NDVI values remained above 0.6, indicating that regreening wood reached high vegetation coverage density after 9 years of regreening. Secondly, in area II, the magnitude of NDVI changes sharply. Within 3 years after regreening, area II showed a trend of first increasing and then decreasing in NDVI values, similar to area I. However, after 3 years, NDVI showed a continuous downward trend, and the overall level remained below 0.5, and only briefly increased during reclamation. Combined with the verification results of interactive interpretation, it was found that area II degenerated three times in 2001, 2010, and 2013, respectively. When regreening was carried out, vegetation improvement was remarkable in a short period of time, but it was difficult to maintain long-term stable conditions for vegetation, and the problem of repeated damage continued to occur. Different from area II, the overall trend of vegetation in area III was better before 2009, and NDVI values remained above 0.5. However, after the damage occurred in 2009, it decreased to the level of 0.4–0.5 and fluctuated continuously, which indicated that the spontaneous combustion of the coal gangue occurred in area III after many years of regreening.

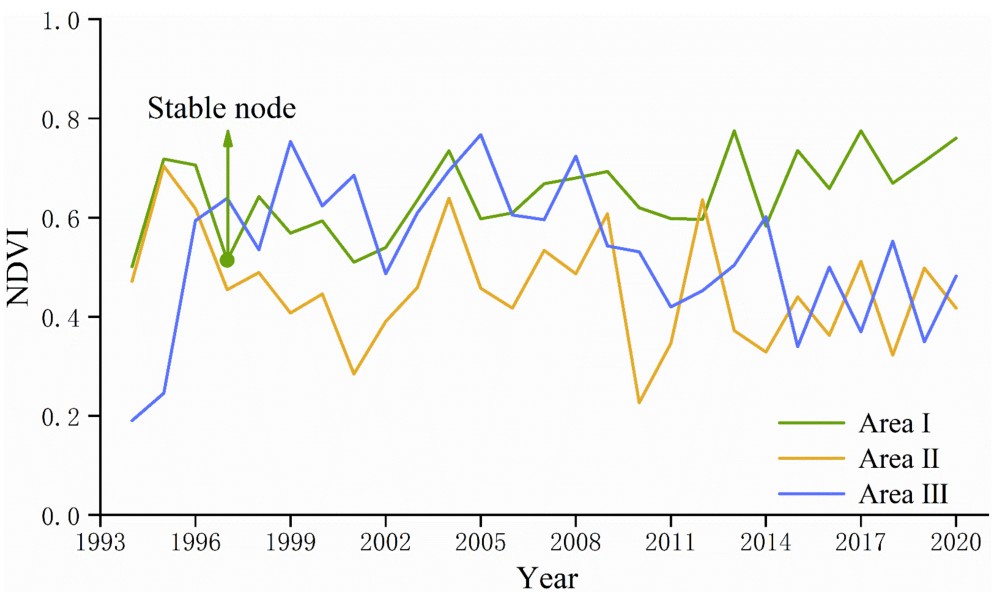

**Figure 4.** Annual average of NDVI in the "typical area" (area I) and the "problem areas" (areas II and III).

In conclusion, under normal recovery conditions, the regreening vegetation with wood as the main cover type took about three years to reach stability. Therefore, when the regreening vegetation type is wood, it is recommended to focus on monitoring after three years of regreening. The time of re-damage is difficult to predict, and may occur 3–4 years after regreening, or even several years after regreening, similar to area II and area III. Due to the spontaneous combustion of coal gangue, it is difficult to restore completely in some mining areas. Timely monitoring for vegetation is helpful for predicting the occurrence of damage.

### 3.2. Monitoring Results in the West Dump

According to the images and historical records, it can be determined that the west dump began to be regreened in 1989. Taking the first regreening time as starting time, unary regression analysis overall and Sen+MK trend analysis on the spatial scale were carried out on the NDVI dataset. From the unary regression analysis results (Figure 5a), NDVI dataset has a significant positive correlation with time, and the correlation coefficient reaches up to 0.95, indicating that NDVI values increased steadily with time. From the Sen+MK trend analysis results (Figure 5b), about 95.58% area of the west dump showed a significant increasing trend from 1989 to 2020, and the area is about 261.98 ha. The remaining areas showed insignificant increasing trends for NDVI, mainly located at the western and southern edges of the dump.

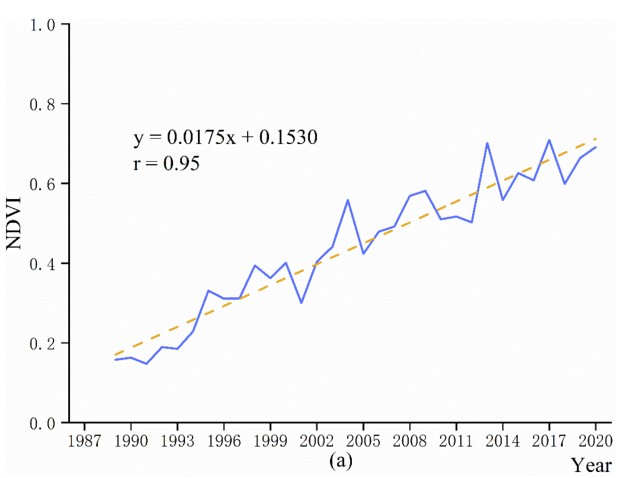

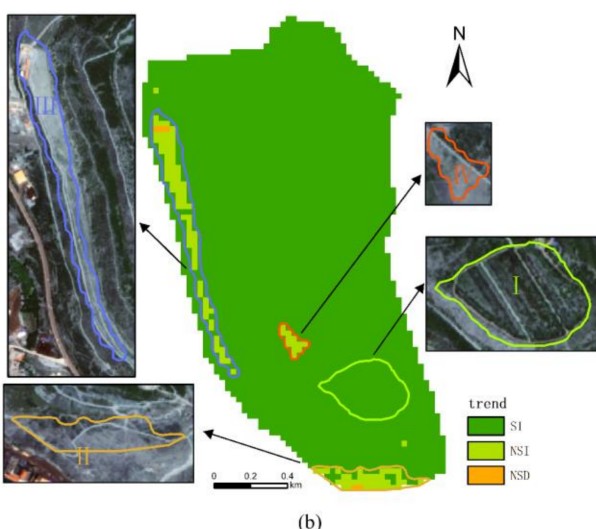

**Figure 5.** The time series analysis for NDVI on time and spatial scales in the west dump. (**a**) shows the results of the unary regression analysis. And (**b**) shows the results of the Sen+MK trend analysis. SI, NSI, NSD, and SD are, respectively, the abbreviations for significantly increase, not significantly increase, not significantly decrease, and significantly decrease.

According to the Sen+MK trend analysis results, the "typical area" (area I) and the "problem areas" (areas II, III, and IV) were extracted, as shown in Figure 5b. Combined with interactive interpretation, it is found that area I, which began to be regreened in 1994, showed a significant increasing trend for NDVI; area II, area III, and area IV were regreened in 1989, 1994, and 1995, respectively. Although area II was the first place to be regreened in the west dump, from the Sen+MK trend analysis results, NDVI showed an insignificant increasing trend for NDVI after long-term restoration, which showed that this area has been greatly disturbed after regreening. Moreover, area III also showed an insignificant trend in vegetation. Moreover, area II and area III are similar in some respects, i.e., they are both located at the edge of the dump, and the regreening time is earlier than in the other areas. The following two reasons will be considered: (1) The area located at the edge of the dump may be affected by road expansion; (2) the early regreening technology is not mature enough, which may affect the later vegetation restoration. In addition, area IV is located inside the dump and is only 1.50 ha, so a focus on examining whether spontaneous combustion of the coal gangue has occurred is necessary.

After extracting the area range based on the Sen+MK trend analysis results, the year-by-year NDVI of each area was obtained by statistical methods and drawn into a line graph, as shown in Figure 6. Firstly, NDVI values in area I increased first and then decreased within 3 years after regreening, and basically reached stability after 3 years. Moreover, based on the Mann–Kendall mutation test, the stable node of area I is about three years. From 1997 to 2012, NDVI values fluctuated from 0.4 to 0.5. After 2012, the values increased significantly and remained at around 0.6, indicating that although regreening species shrub basically reaches a stable state in 3 years, higher vegetation coverage requires more time. Then in area II, the NDVI values were lower than 0.5 before 2002, and there was a cliff-like decline in 1993. After 2002, NDVI values fluctuated in the range of 0.4 to 0.6, but the fluctuation range was large. Similarly, NDVI values of the area located at the western edge also showed continuous fluctuations. Moreover, area III degraded several times in 2001, 2005, and 2013, respectively. Verified by interactive interpretation, degradation did occur in the corresponding years. Last, in area IV, NDVI values experienced a massive decline in 2007 and then recovered again in 2009. Judging from the trend of NDVI after regreening, the effect of the second regreening was relatively good, and the vegetation growth state did not fluctuate violently and remained relatively stable.

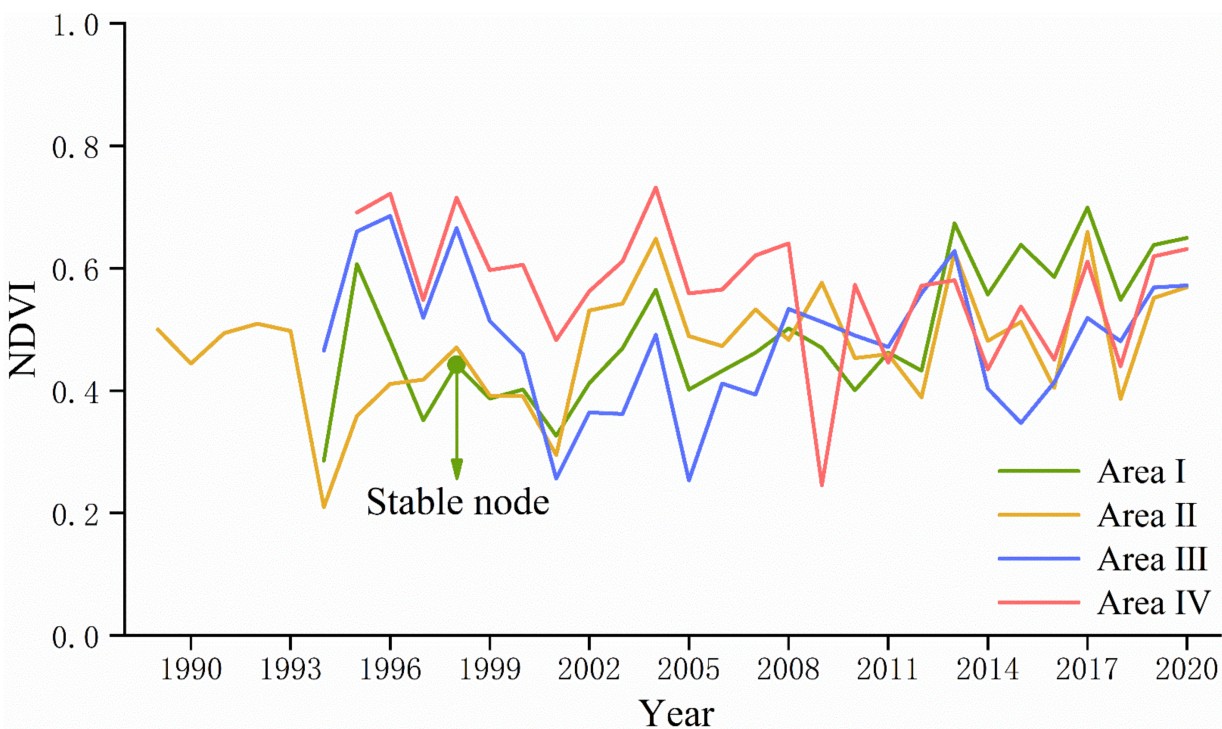

**Figure 6.** Annual average of NDVI in the "typical area" (area I) and the "problem areas" (areas II, III, and IV).

In conclusion, under normal recovery conditions, the regreening vegetation with shrubs as the main cover type took about three years to reach stability. Moreover, in the west dump, the regreening vegetation at the dump edge is easily disturbed by other factors, resulting in drastic changes.

### 3.3. Monitoring Results in the West Expansion Dump

According to the image information and historical data, the regreening project in the west expansion dump started in 2003. The total area of the west expansion dump is about 351.9589 ha, and the regreening time is longer. Taking the first regreening time as starting time, unary regression analysis overall and Sen+MK trend analysis on the spatial scale were carried out on the NDVI dataset. From the regression analysis results (Figure 7a), NDVI values showed an overall increasing trend from 2003. In addition, NDVI has a strong

correlation with time, and the Pearson correlation coefficient reaches about 0.92. From the perspective of trend analysis (Figure 7b), the results showed that the area with a significant increasing trend in NDVI accounted for 79.89% of the total dump, and the area with an insignificant increasing trend accounted for 18.19%. In addition, the areas where the trend was not significant were concentrated and contiguous in the interior of the dump and scattered at the edge of the dump.

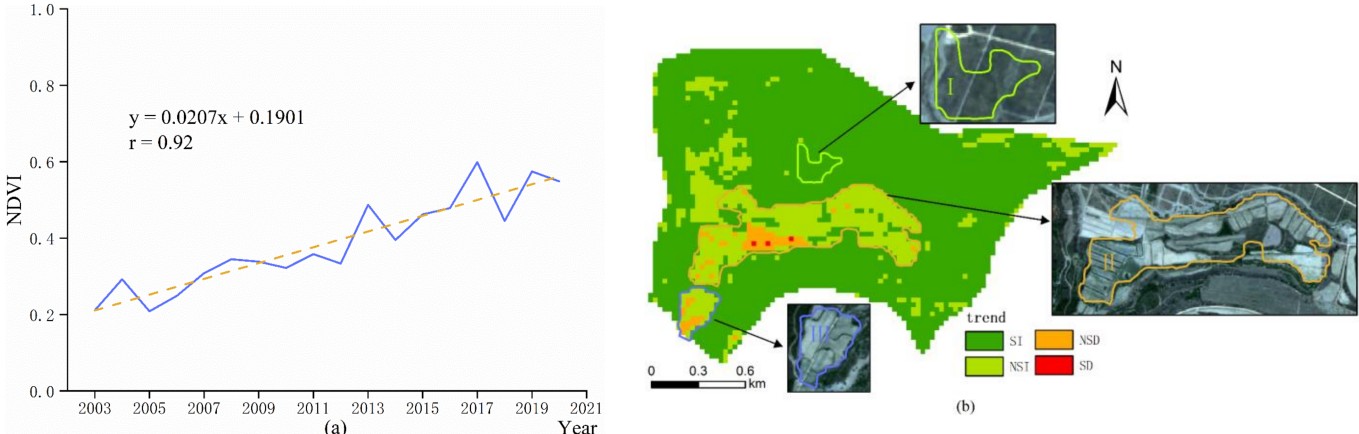

**Figure 7.** The time series analysis for NDVI on time and spatial scales in the west expansion dump. (**a**) shows the results of the unary regression analysis. And (**b**) shows the results of the Sen+MK trend analysis. SI, NSI, NSD, and SD are, respectively, the abbreviations for significantly increase, not significantly increase, not significantly decrease, and significantly decrease.

According to the Sen+MK trend analysis results, the following "typical area" (area I) and the "problem areas" (areas II and III) were extracted. According to Figure 7b, area I is in the middle of the dump and started to be regreened in 2010. After regreening, the improvement effect of vegetation is remarkable, indicating that there was less disturbance during restoration. Moreover, area II and area III have insignificant increasing trends for NDVI values. Moreover, the reason why other areas with insignificant trends were not extracted was that the regreening time was less than 8 years, which has little significance for research reference.

After extracting the area range based on the Sen+MK trend analysis results, the year-by-year NDVI of each area was obtained by statistical methods and drawn into a line graph, as shown in Figure 8. NDVI values in the three areas all showed an upward trend, but the improvement effect in area I is more prominent than in the other two areas. According to the results of the Mann-Kendall mutation test, the mutation node was one year, indicating that the grass reached stability in one year. It shows that it takes less time for grass to stabilize compared to other vegetation types. Moreover, according to the field investigation, the main vegetation coverage type is pasture in area II. Therefore, it is speculated that low NDVI values are caused by the harvest of forage from August to October. Moreover, through on-site investigation, we found that area III collapsed after many years of regreening, and a land leveling project was carried out.

In conclusion, under normal recovery conditions, the vegetation with grass as the main cover type reaches a stable time node of about one year. Moreover, after regreening of the dump, vegetation may improve significantly in a short period of time, but low vegetation coverage may occur due to unsuitable regreening seedlings or later surface subsidence.

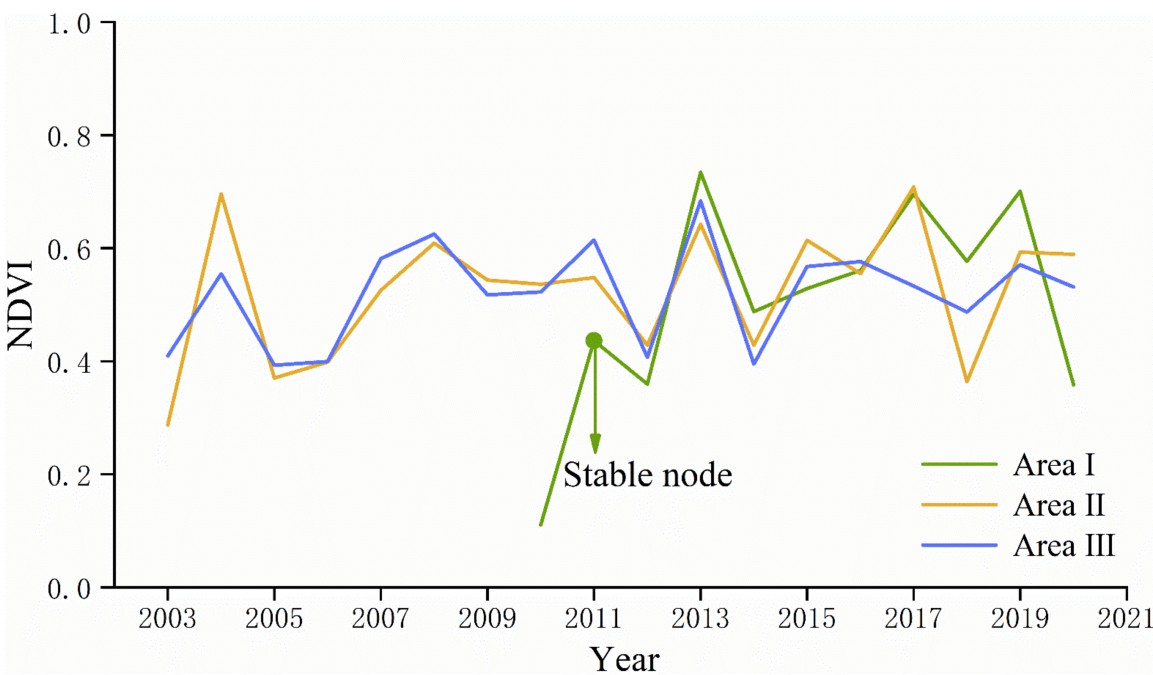

**Figure 8.** Annual average of NDVI in the "typical area" (area I) and the "problem areas" (areas II and III).

### 3.4. Monitoring Results in the Inner Dump

According to the image information and historical data, the disposal of the inner dump began in 1987, and the reclamation began in 1998. To date, the inner dump has completed all disposal work, but the reclamation is still going on. Taking the first regreening time as starting time, unary regression analysis overall and Sen+MK trend analysis on the spatial scale were carried out on the NDVI dataset. From the perspective of regression analysis (Figure 9a), the positive correlation between NDVI and time was low from 1998 to 2020, and the Pearson correlation coefficient was only 0.50. Moreover, in the ten years from 2002 to 2012, the variation range of NDVI values was small, and remained around 0.2. From Sen+MK trend analysis results (Figure 9b), the areas with significantly improved vegetation accounted for the largest proportion in the inner dump, about 86.56%. According to the actual situation, the type of land use was changed later, including the construction of artificial lakes and buildings, which affected the vegetation growth trend in the central area.

According to the Sen+MK trend analysis results, the "typical area" (area I) and the "problem areas" (areas II and III) were extracted, as shown in Figure 9b. Combined with the results of interactive interpretation, area I, area II and area III were regreened in 2003, 1998 and 2004 respectively. Through field investigation, it is found that area II is close to industrial sites and is greatly affected by human activities.

After extracting the area range based on the Sen+MK trend analysis results, the year-by-year NDVI of each area was obtained by statistical methods and drawn into a line graph. According to Figure 10, the trends of the three areas are quite different. Firstly, area I showed a significant increasing trend in NDVI values overall. According to the Mann–Kendall mutation test results, the time for area I reaching stability was three years. Within three years after regreening, NDVI values changed greatly, showing a trend of first increasing and then decreasing. After that, although NDVI still showed a phenomenon of continuous fluctuation, the band amplitude was always relatively small and basically reached a stable state. Secondly, the fluctuation of NDVI in area II was small after regreening, but the improvement effect of vegetation was not obvious after a long period of time. This is due to the human activities in the southern industrial site, which have caused great disturbance to the ecology of area II. Moreover, area III also showed a steady increasing trend after regreening, but after the cliff-like decline in 2010, the NDVI was always less than 0.1, maintaining a low level. Combined with interactive interpretation, it is found that in 2010,

the land use type of area III was changed, and locals built new farms to increase social benefits. So, the vegetation coverage was very low later.

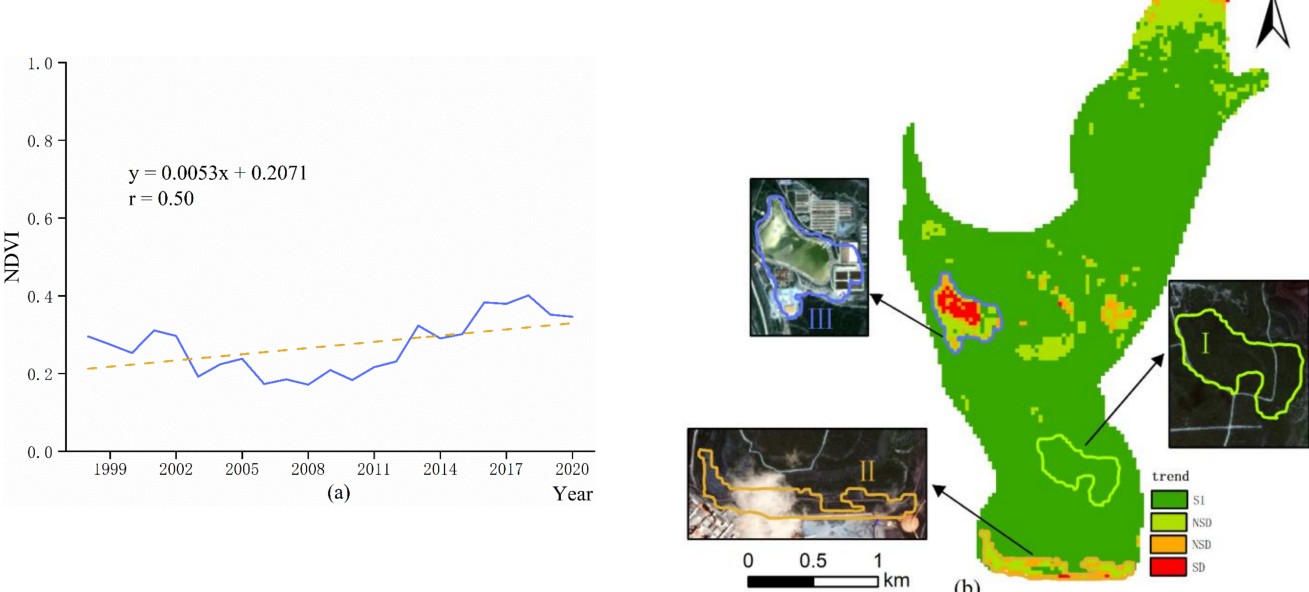

**Figure 9.** The time series analysis for NDVI on time and spatial scales in the inner dump. (**a**) shows the results of the unary regression analysis. And (**b**) shows the results of the Sen+MK trend analysis. SI, NSI, NSD, and SD are, respectively, the abbreviations for significantly increased, not significantly increased, not significantly decreased, and significantly decreased.

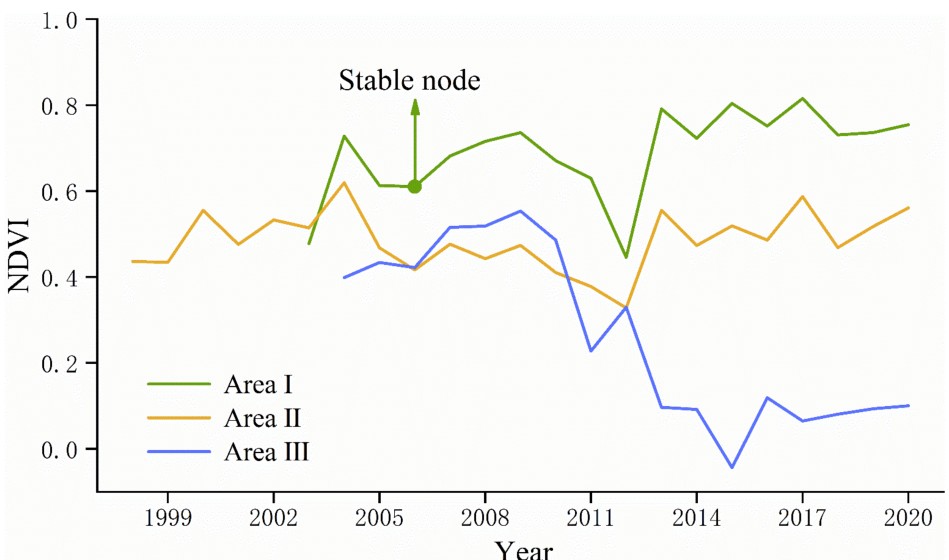

**Figure 10.** Annual average of NDVI in the "typical area" (area I) and the "problem areas" (areas II and III).

In conclusion, the inner dump reached a stable time node of about three years under normal recovery. Regreening vegetation is susceptible to human disturbance, resulting in low vegetation density.

## 4. Discussion

### 4.1. The Influence of Spontaneous Combustion of Coal Cangue on Vegetation

The degenerate expansion is most likely caused by the spontaneous combustion of the gangue at the bottom. This is because the spontaneous combustion of the coal gangue will not only directly affect the growth of the upper surface plants but also cause the surrounding soil temperature to rise, thereby affecting the growth of surrounding vegetation. Through the Sen+MK trend analysis in this study, it was detected that there was an area where coal gangue spontaneously combusted in the east of the south dump. Moreover, through the annual average curve of NDVI, it is found that the spontaneous combustion of the coal gangue is repeatable.

In the early stage of mining restoration, due to the immature technology and purpose of saving engineering volume, the topsoil is often directly overlaid on top of the coal gangue, then the regreening project was carried out on the topsoil. However, after years of monitoring, it has been found that the pyrite, residual coal, and other combustibles attached to the bottom coal gangue will undergo a slow oxidation reaction. When meeting suitable heat storage conditions, adsorption heat and the heat released by oxidation will make the temperature of the coal gangue raise slowly. When the temperature reaches the ignition point, the coal gangue would burn, and the porous nature of the coal gangue and the diffusion of gas molecules ensure that there is a sufficient oxygen supply. In the process of spontaneous combustion of coal gangue, not only will the upper vegetation be directly burned, but also a huge number of harmful gases and soot will be emitted, which will cause serious pollution to the surrounding environment [51–54]. More seriously, if rainfall is heavy, a relatively strong leaching effect will occur, and water vapor may explode under heating conditions.

The problem of spontaneous combustion of the coal gangue has always been a major problem in mine restoration [55]. In recent years, relevant scholars have been committed to exploring the monitoring and repair of the coal gangue spontaneous combustion. Ruan, M et al. used drones, field surveys, and indoor analysis to analyze the surface temperature, vegetation coverage, and soil nutrients of gangue piles with different degrees of spontaneous combustion, and found that high-temperature stress affected plant survival [56].

### 4.2. The Influence of Terrain on Vegetation

Open-pit coal mining will destroy the original landform and landscape, the stacking of stripped materials completely changed the original landform with vertical and horizontal ravines, forming a stepped landform with slopes and large platforms. Topographic design and silviculture technology were both needed before revegetation [57]. Landform remodeling is the basis for future land use in mining areas, and it is also a key research issue for land reclamation and ecological reconstruction. Some scholars indicated that there is interaction and feedback between terrain and vegetation dynamics [58,59]. Vegetation dynamics are largely affected by terrain factors such as elevation, slope, and slope aspect. At the same time, dense vegetation coverage will greatly reduce the risk of soil erosion.

However, due to the influence of precipitation and other factors, accidents such as landslides and subsidence often occur on the slopes, resulting in the decrease in the thickness of the effective soil layer. Relevant studies have shown that the regreening vegetation on the slopes is more susceptible to the impact of high precipitation than that on the platforms. On the slopes, reclamation projects lack the soil and water environment required for plant growth and the conditions for plants to firmly climb, so it has always been a difficult technical problem in the construction of mine ecological restoration [60–62].In recent years, some scholars have comprehensively used hydrology and geomorphology in land reclamation and ecological reconstruction in mining areas, and have continuously advanced the theoretical system of topographic remodeling to maturity [63–65]. With the continuous improvement of geological disaster management technology, terrain remodeling

will become an indispensable and most important basic part of the ecological reconstruction of abandoned mining areas.

### 4.3. The Role of Vegetation in Soil Erosion

The relationship between vegetation and soil erosion has been an important part of eco-environmental research [66,67]. As an important factor affecting soil erosion, the control function of vegetation on soil erosion is mainly embodied in the loss of dynamic energy of rainfall [68,69]. Most research has concluded that improving vegetation cover is an effective method for controlling soil erosion and that the effectiveness of erosion control varies by vegetation type. Juanzhu Liang pointed out that there was a significant correlation between rainfall and soil erosion. The amount of erosion increased with the increase in rainfall. In addition, differences in the response of different vegetation covers to rainfall were evident, with bare soil and tree areas showing a high response to changes in soil erosion to rainfall, while herbaceous areas with high cover showed a less pronounced response to rainfall [70]. Therefore, effective erosion control requires first a reasonable choice of plant species and then a reasonable spatial layout of the vegetation. This step is the key to linking soil reconstruction and biodiversity reorganization in mining reclamation. Taking the reclamation of the dumps in the Antaibao mine as an example, the combination of grass and irrigation is adopted in the reclamation of the slopes [71,72]. While increasing the vegetation cover by planting grass, the soil structure is stabilized by the plant roots of shrubs, such as sea buckthorn (*Hippophae rhamnoides* Linn.) and Caragana (*Caragana korshinskii* Kom). *Moreover,* in this study, the results showed that both the shrubs in the west dump and the grasses in the west expansion dump steadily improved over time, with a strong positive correlation with time. Therefore, it is recommended that in the early stages of vegetation restoration for mining areas, the combination of "grass and shrub" is used to regreen. This pattern of revegetation ensures an increase in vegetation cover while stabilizing the soil structure.

However, previous studies have mainly concentrated on the control function of vegetation on soil erosion, while there are fewer studies on how erosion affects vegetation, and more research is needed. Understanding the feedback mechanisms of both can contribute to vegetation restoration and erosion control. In addition, the interactive processes of vegetation and soil erosion modify the microtopography. In turn, the changes in microtopography further influence vegetation and erosion patterns. The mechanism of this process should also be intensively researched and explored.

### 4.4. Limitations of Interactive Interpretation

When interactive interpretation was performed in this study, NDVI values of the previous year were used as the benchmark, and the time point and scope of the degraded area were determined by comparing them with the NDVI values of the study year. However, this process was inevitably affected by the subjective judgment of readers, which affects the interpretation results. When different readers interpret the images, there are certain differences in results [73]. Although interactive interpretation is widely used in the processing of remote sensing images, there are many factors that affect interactive interpretation, which leads to some errors [74,75]. Related studies have proposed some methods to reduce the error of interactive interpretation, for example, when detecting certain types of land changes, the consistent date of image collection is important, and images in the same season can upgrade the accuracy of interpretation [76].

## 5. Conclusions

In this study, the NDVI time series dataset from 1986 to 2020 was obtained by Landsat images and the HJ image. Moreover, through the unary regression analysis overall and Sen+MK trend analysis on the spatial scale, the growth trends of regreening vegetation in each dump were obtained. Combined with interactive interpretation and the Mann–Kendall mutation test, information such as stable nodes of different regreening vegetation and vegetation growth patterns in degraded areas were obtained. The conclusions are as follows:

(1) After regreening, NDVI values all showed increasing trends within a short period. Moreover, due to the distinction in the reclamation mode, the growth trends of regreening vegetation in each dump showed certain regularity. The main performance is: the earlier the regreening time, the more areas that are covered by significantly improved vegetation. In this study: 97.31% (the proportion of significantly improved vegetation in the south dump) > 95.58% (the proportion in the west dump) > 86.56% (the proportion in the inner dump) > 79.89% (the proportion in the west expansion dump).

(2) Different types of regreening vegetation have different time points for reaching stability. In this study, by extracting the "typical area" with significantly increasing trends in NDVI values for the Mann–Kendall mutation test, it takes about three years for wood, shrub, and a mix of grass, and shrub and wood to reach stability, but only one year for grass.

(3) The degraded areas in the mining area were expansive and repetitive. Repeatability means that the degraded area was very likely to degrade once more after the second revival. For example, area III in the west dump was damaged in 2000 and 2013, respectively. Expansion means that the degraded area may extend to the surroundings in the next degradation, similar to area III in the south dump.

**Author Contributions:** Conceptualization, J.H. and B.Y.; methodology, J.H. and B.Y.; formal analysis, J.H.; investigation, J.H. and B.Y.; resources, J.H. and B.Y.; data curation, J.H.; writing—original draft preparation, J.H.; writing—review and editing, Z.B., B.Y., and Y.F.; supervision, Z.B. and B.Y.; project administration, Z.B.; funding acquisition, Z.B. All authors have read and agreed to the published version of the manuscript.

**Funding:** This research was funded by the National Natural Science Foundation of China: Effects of land reclamation on changes and trade-offs of ecosystem services in mining areas (42007425).

**Data Availability Statement:** The remote sensing data used in this article are public and can be found at https://earthexplorer.usgs.gov/ (accessed on 14 July 2021) and http://www.cresda.com/CN/ (accessed on 22 July 2021).

**Acknowledgments:** We express our gratitude to the anonymous reviewers and editors for their professional comments and suggestions.

**Conflicts of Interest:** The authors declare no conflict of interest.

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
