# Peer review of "Remote Sensing Monitoring of Vegetation Reclamation in the Antaibao Open-Pit Mine"

_remotesensing, doi:10.3390/rs14225634_

Round 1

Reviewer 1 Report (Previous Reviewer 3)

I think you have dealt with the suggestions I made. Good work.

Author Response

Thank you very much for your recognition of our manuscript.

Reviewer 2 Report (Previous Reviewer 4)

Attached.

Author Response

We appreciate your kind suggestions and we have amended the manuscript accordingly.

Reviewer 3 Report (Previous Reviewer 5)

Accept

.

Author Response

Thank you very much for your recognition of our manuscript.

This manuscript is a resubmission of an earlier submission. The following is a list of the peer review reports and author responses from that submission.

Round 1

Reviewer 1 Report

Hi.

I added some comments to the manuscript. Please find them attached. 

Bests, 

Reviewer

Author Response

Thank you very much for your suggestion, I have made the changes. The modifications are as in the attachment.

Reviewer 2 Report

THE PAPER NEEDS TO IMPROVE THE GRAPHS AND IMPROVE THE DISCUSSION... WE NEED TO SEE THIS IMPROVEMENTS

Author Response

(The authors gave the same response as above.)

Reviewer 3 Report

The subject is relevant in a world facing climate change and coal burning being one of the causes of global warming.  Some parts of the manuscript have to be clarified and information added. Did you perform a field work? Why did you combine 2 statistical trend analysis techniques? I am sending you the pdf file with notes. I suggest a new title: Monitoring regreen of vegetation in a Chinese mining area using remote sensing

Author Response

(The authors gave the same response as above.)

Reviewer 4 Report

Dear authors, 

Please, see the attached document with my comments.

Cheers.

Author Response

(The authors gave the same response as above.)

Reviewer 5 Report

Dear Authors,

The manuscript prepared by you contains valuable datasets that deserve to be published and would be of interest for decision makers and land managers all over the world. However, it is suggested to take into consideration the specific comments provided below to improve overall quality of the manuscript.

 Specific comments:

1 - There is a lack of the brief description of the background and significance of the study in the abstract.

2 - Please describe more in detail about the nonparametric statistical methods and analyze the trend of time series data in the section introduction.

3 - Please provide more detail about the land use/ land cover of the study area.

4- The figures in the article need to be optimized by adjusting the font and color.

5 - The discussion in the article is needs to be improved. It is suggested to propose possible sustainable soil management strategies based on the analysis of the existing research results.

Author Response

(The authors gave the same response as above.)
